# META-LEARNING A DYNAMICAL LANGUAGE MODEL

**Thomas Wolf, Julien Chaumond & Clement Delangue**
Hugging Face Inc.
81 Prospect St.
Brooklyn, New York 11201, USA
`{thomas,julien,clement}@huggingface.co`

## 1 INTRODUCTION

Language modeling is a prototypical unsupervised task of natural language processing (NLP). It has triggered the development of essential bricks of models used in speech recognition, translation or summarization. More recently, language modeling has been shown to give a sensible loss function for learning high-quality unsupervised representations in tasks like text classification (Howard & Ruder, 2018), sentiment detection (Radford et al., 2017) or word vector learning (Peters et al., 2018) and there is thus a revived interest in developing better language models. More generally, improvement in sequential prediction models are believed to be beneficial for a wide range of applications like model-based planning or reinforcement learning whose models have to encode some form of memory.

One of the main issues limiting the performance of language models (LMs) is related to capturing long-term dependencies within a sequence. Neural network based language models (Hochreiter & Schmidhuber, 1997; Cho et al., 2014) learn to implicitly store dependencies in a vector of hidden activities (Mikolov et al., 2010). They can be extended by attention mechanisms or memories/caches (Bahdanau et al., 2014; Tran et al., 2016; Graves et al., 2014) to capture long-range connections more explicitly. Unfortunately, the very local context is often so highly informative that LMs typically end up using their memories mostly to store short term context (Daniluk et al., 2016).

In this work, we study the possibility of combining short-term representations stored in hidden states with medium term representations encoded in a set of dynamical weights of the language model. Our work extends a series of recent experiments on networks with dynamically evolving weights (Ba et al., 2016; Ha et al., 2016; Krause et al., 2017) which shows improvements in sequential prediction tasks. We build upon these works by formulating the task as a hierarchical online meta-learning task as detailed below.

The motivation behind this work stems from two observations. First, there is evidence from a physiological point of view that time-coherent processes like working memory can involve differing mechanisms at differing time-scales. Biological neural activations typically have a 10ms coherence while short-term synaptic plasticity operates on longer timescales of 100ms to minutes (Tsodyks et al., 1998; Ba et al., 2016), before long-term learning kicks in at longer time scales of a few minutes. Second, multiple time-scales dependencies in sequential data can naturally be encoded by using a hierarchical representation where higher-level features are changing slower than lower-level features (Schmidhuber, 1992; Chung et al., 2016).

As a consequence, we would like our model to store information in a multi-scale hierarchical way where

1. *short time-scale representations* can be encoded in neural activations (hidden state),
2. *medium time-scale representations* can be encoded in the dynamic of the activations by using dynamic weights, and
3. *long time-scale memory* can be encoded in a static set of weights of the network.

In the present work, we take as starting point an RNN language model, and associate to each weight $\theta$ a dynamic weight $\hat{\theta}$ of identical shape. The sum of the static and dynamic weights $\theta + \hat{\theta}$ is used in the language model, in place of the original weight $\theta$, to compute the output. The resulting weights can thus be seen as online trained weights as we now detail further.

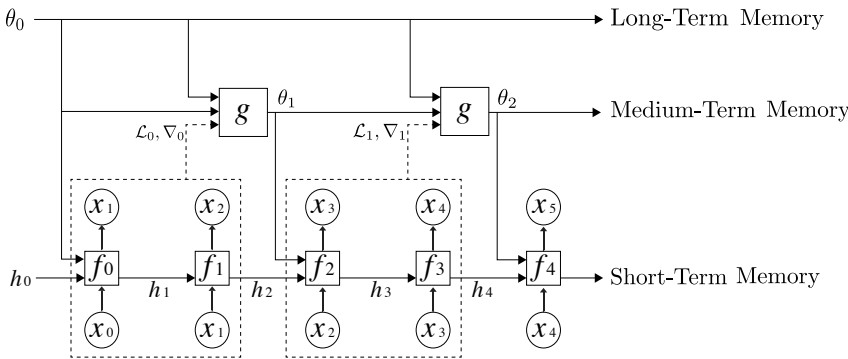

Figure 1: A diagram of the Dynamical Language Model.

## 2 DYNAMICAL LANGUAGE MODELING

Given a sequence of $T$ discrete symbols $S = (w_1, w_2, \ldots, w_T)$, the language modeling task consists in assigning a probability $P(S) = p(w_1, \ldots, w_T)$ to the sequence. $P(S)$ can be written as

$$P(S \mid \theta) = \prod_{t=1}^{T} P(w_t \mid w_{t-1}, \ldots, w_0, \theta) P(w_0 \mid \theta). \tag{1}$$

In the case of a neural-network-based LM, the conditional probability $P(w_t \mid w_{t-1}, \ldots, w_0, \theta)$ is typically parametrized using an autoregressive neural network as $P(w_t \mid w_{t-1}, \ldots, w_0, \theta) = f_\theta(w_{t-1}, \ldots, w_0)$, $\theta$ being a set of parameters of the language model network.

In the dynamical framework, the parameters $\theta$ of the language model are not tied over the sequence $S$ but are allowed to evolve. Prior to computing the probability of a future token $w_t$, a set of parameters $\theta_t$ is estimated from the past parameters and tokens as $\theta_t = \underset{\theta}{\mathrm{argmax}}\, P(\theta \mid w_{t-1}, \ldots, w_0, \theta_{t-1} \ldots \theta_0)$ and the updated parameters $\theta_t$ are used to compute the probability of the next token $w_t$.

In the hierarchical model, the updated parameters $\theta_t$ are estimated by a higher level neural network:

$$\theta_t = g_\phi(w_{t-1}, \ldots, w_0, \theta_{t-1} \ldots \theta_0) \tag{2}$$

where $\phi$ is the set of (static) parameters of the higher level network (meta-learner).

### 2.1 META-LEARNING FORMULATION

The function computed by the higher level network $g$, estimating $\theta_t$ from an history of parameters $\theta_{<t}$ and data points $w_{<t}$, can be seen as an online meta-learning task in which the higher level network is a meta-learner trained to learn an update rule for the weights of the lower-level network that generalizes an (online) gradient descent rule $\theta_t = \theta_{t-1} - \alpha_t \nabla_{\theta_{t-1}} \mathcal{L}_t$ (where $\alpha_t$ is a learning rate at time $t$ and $\nabla_{\theta_{t-1}} \mathcal{L}_t^{LM}$ is the gradient of the loss of the LM on the $t$-th dataset with respect to previous parameters $\theta_{t-1}$).

Ravi & Larochelle (2016) made the observation that a gradient descent rule bears similarities with the update rule for LSTM cell-states $c_t = f_t \odot c_{t-1} + i_t \odot \tilde{c}_t$ when $c_t \to \theta_t$, $i_t \to \alpha_t$ and $\tilde{c}_t \to -\nabla_{\theta_{t-1}} \mathcal{L}_t$

We extend this analogy to a hierarchical recurrent model, illustrated on figure 1, with:

1. *Lower-level / short time-scale*: a RNN-based language model $f$ encoding representations in the activations of a hidden state,

2. *Middle-level / medium time-scale*: a meta-learner $g$ updating the set of weights of the language model to store medium time-scale representations, and

3. *Higher-level / long time-scale*: a static long-term memory of the dynamic of the RNN-based language model (see below and appendix A).

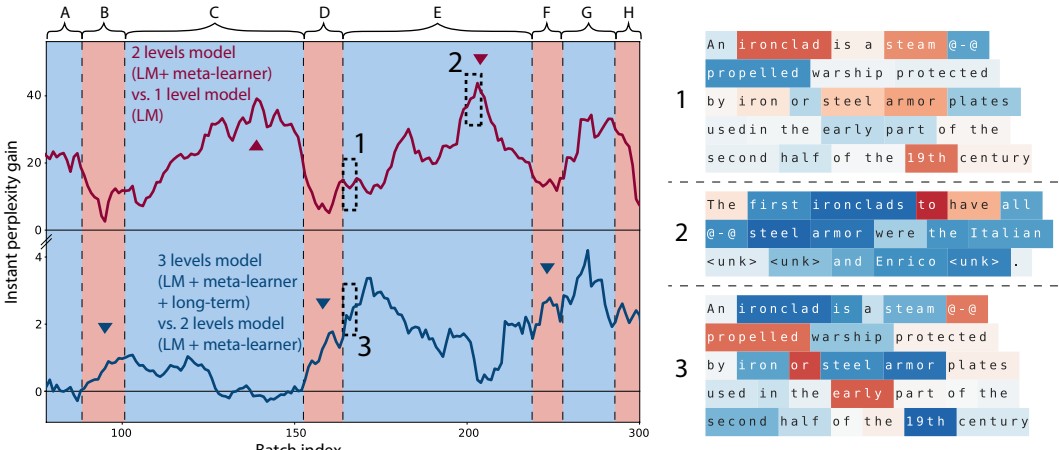

Figure 2: Effect of the medium and long term memories on a sample of Wikitext-2 test set composed of a sequence of Wikipedia articles (letters $A - H$). (Left) Instantaneous perplexity gain: difference in batch perplexity between models. Higher values means the first model has locally a lower perplexity than the second model. (Top curve) Comparing one-level model (LM) with two-levels model (LM + meta-learner). The meta-learner is able to learn medium-term representations so that perplexity is progressively reduced along articles (see C and E, and right samples 1 and 2). (Bottom curve) Comparing two-levels model with three-levels model (meta-learner + LM + long-term memory). Static long-term memory reduce the forgetting of the pre-trained LM task. Perplexity is reduced at topics changes and beginning of new topics (see B, D and F and right sample 3 versus 1). (Right) Token loss difference on batch samples indicated on the left curves. Blue means the first model has a lower token loss than the second model, red means higher. (1 vs 2) The two-level model learn to remember "ironclad" and "steel armor" (3 vs 1). The static long-term memory improves the loss at the beginning of a new topic for common words like "steel armor" and "19th century".

The meta-learner $g$ is thus trained to update the lower-level network $f$ by computing $f_t, i_t, z_t = g_\phi(\theta_{t-1}, \mathcal{L}_t^{LM}, \nabla_{\theta_{t-1}} \mathcal{L}_t^{LM}, \theta_0)$ and updating the set of weights as

$$\theta_t = f_t \odot \theta_{t-1} + i_t \odot \nabla_{\theta_{t-1}} \mathcal{L}_t^{LM} + z_t \odot \theta_0 \qquad (3)$$

In analogy with a hierarchical recurrent neural networks (Chung et al., 2016), the gates $f_t, i_t$ and $z_t$ can be seen as controlling COPY, FLUSH and UPDATE operations:

1. COPY ($f_t$): part of the state copied from the previous state $\theta_{t-1}$,

2. UPDATE ($i_t$): part of the state updated by the loss gradients on the previous batch, and

3. FLUSH ($z_t$): part of the state reset from the static long term memory $\theta_0$.

## 3 EXPERIMENTS

We performed experiments on the Wikitext-2 dataset Merity et al. (2016) using an AWD-LSTM LM (Merity et al., 2017) and a feed-forward meta-learner. Test perplexities are similar to perplexities obtained using dynamical evaluation (Krause et al., 2017), reaching $46.9$ when starting from a pre-trained LM test perplexity of $64.8$. A few quantitative experiments are illustrated on Figure 2.

The hierarchical model is trained in two steps. First a set of static weights $\theta_0$ is obtained by performing elastic weight consolidation (Kirkpatrick et al., 2017) as described in appendix A. Then, the meta-learner $g$ is trained in an online meta-learning fashion: a training sequence $S$ is split in a sequence of mini-batches $B_i$, each batch $B_i$ containing $M$ inputs tokens $(w_{i \times M}, \ldots, w_{i \times M + M})$ and $M$ associated targets $(w_{i \times M + 1}, \ldots, w_{i \times M + M + 1})$. The meta-learner is trained as described in (Andrychowicz et al., 2016; Li & Malik, 2016) by minimizing the sum over the sequence of LM losses: $\mathcal{L}_{meta} = \sum_{i>0} \mathcal{L}_i^{LM}$. More details on the training process are given in Appendix B.

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

## A  ELASTIC WEIGHTS CONSOLIDATION

In this work, we assume that current state-of-the-art LMs like the AWD-LSTM of Merity et al. (2017) are modeling short time-scale dependencies in a satisfactory way [1]. Since the meta-learner (medium level of the model of figure 1) implements a form of continual-learning, the hierarchical model faces the phenomenon known as catastrophic forgetting (French, 1999) which occurs when a network is trained sequentially on multiple tasks and the weights that are important for a previous task are changed to meet the objectives of a more recent task.

To reduce this effect we add the higher-level static memory and initialize it by using "elastic weight consolidation" that was introduced by Kirkpatrick et al. (2017) to prevent catastrophic forgetting in multi-task reinforcement learning. The static set of weights $\theta_0$ of the higher-level static memory is thus computed as a Laplace approximation to the posterior distribution $p(\theta \mid \mathcal{D})$ of the pre-trained weights with mean given by the set of pre-trained parameters and a diagonal precision given by the diagonal of the Fisher information matrix $F$ (which can be computed from the first-order derivative of the LM loss function $\mathcal{L}^{LM}$).

## B  TRAINING THE ONLINE META-LEARNER

The meta-learner is trained by truncated back-propagation through time and is unrolled over at least 40 steps as the reward from the medium-term memory is relatively sparse Li & Malik (2016). To be able to unroll the model over a sufficient number of steps while using a state-of-the-art language model with about 20-30 millions parameters, we use a memory-efficient version of back propagation through time with gradient checkpointing (Gruslys et al., 2016). The meta-learner is a simple feed-forward network which implement coordinate-sharing as described in Andrychowicz et al. (2016); Ravi & Larochelle (2016).

---

[1] Even though there may of course still be room for improvement

