# OpenReview forum: "Meta-Learning a Dynamical Language Model"
_ICLR.cc/2018/Workshop — Accept_

### Official Review · AnonReviewer3 · 2018-03-09
**Meta-learning but for what?**

**Rating:** 6
**Confidence:** 3

**Review:**

The paper explores how to incorporate test time information into a trained language model.

Previous works include the Neural Cache and Dynamic Evaluation. The latter keeps taking small gradient descent steps while working through the test set. The authors recast Dynamic Evaluation as Meta-Learning where the weights of a pretrained RNN are updated by a meta model which is another RNN. The meta model has access to the loss and to the gradients of the loss with respect to the weights, plus the pretrained weights.

Having the same information available as the cited Krause et al. 2017 paper, they proceed to get the same results too, but in a meta learning formulation.

Pros:
- The formulation of the meta learning model is interesting.
- The view of the overall model as having memories of three different time scales is intriguing.
- Gorgeous and informative visualization of the contribution of the meta model.

Cons:
- It's not very novel: it's a reformulation with meta learning (which was proposed before).
- The meta learning doesn't seem to contribute anything.

Maybe this could be better presented as a negative result?

---

### Official Review · AnonReviewer2 · 2018-03-11

**Rating:** 5
**Confidence:** 3

**Review:**

== summary ==
The authors describe a new technique to perform dynamic evaluation of recurrent neural network language models. The proposed method is inspired by "fast weights", more specifically, using an additional network to update the weights of the models to adapt to change in the data. Here, the authors propose to use the method introduced by Ravi and Larochelle: each new weight is a linear combination of the previous weight, the gradient, and the weight of the static model. The weights of this linear combination are computed by the additional network. This method can be seen as an extension of stochastic gradient descent, where the learning rate for each weight is computed by a network. The authors evaluate the method on the WikiText-2 dataset.

== assessment ==
This paper is clearly written and easy to follow (and explain connection between dynamic evaluation and meta learning). However, I believe that the contributions are a bit incremental. The main idea of the paper is to combine dynamic evaluation of RNNLM (where SGD is used to update the model) and the meta learning approach of Ravi and Larochelle. The experimental results are a bit disappointing, since the proposed method obtains similar results as SGD (SGD: 44.3, here 46.9), with SGD being a special case of the model.

== pros and cons ==
+ well written paper, connection between dynamic eval and meta learning.
- a bit incremental
- weak experimental results

---

### Official Review · AnonReviewer1 · 2018-03-13
**Meta learning for dynamic language models**

**Rating:** 8
**Confidence:** 4

**Review:**

This paper uses meta learning to learn medium time-scale representations in dynamic language models, which is potentially useful for language modeling on long texts where contextual information transmitted over long distances can be beneficial.

On language modeling, they perform at par with previous work in dynamic language models, and their qualitative analysis suggest that their dynamic language model is able to learn medium-term representations.

This is an interesting paper and a promising line of work.

---

### Decision · Program_Chairs · 2018-03-20
**ICLR 2018 Workshop Acceptance Decision**

**Decision:**

Accept

**Comment:**

Congratulations, your paper was accepted to the ICLR workshop.